# Toward National Guidelines for Biodegradable and Compostable Bioplastics: A Case Study in the Federal Territory of Kuala Lumpur, Malaysia

**DOI:** 10.3390/polym17162165

**Published:** 2025-08-08

**Authors:** Zurina Mahadi, Emirul Adzhar Yahya, Mashitoh Yaacob, Wardah Mustafa Din, Ahmad Firdhaus Arham, Nur Asmadayana Hasim

**Affiliations:** 1Pusat Pengajian Citra Universiti, Universiti Kebangsaan Malaysia, Bangi 43600, Malaysia; kina@ukm.edu.my (Z.M.); mash@ukm.edu.my (M.Y.); wardahmustafadin@ukm.edu.my (W.M.D.); benferdaoz@ukm.edu.my (A.F.A.); 2Institut Islam Hadhari, Universiti Kebangsaan Malaysia, Bangi 43600, Malaysia

**Keywords:** biodegradable and compostable bioplastics, guidelines, sustainability

## Abstract

Malaysia has committed to phasing out single-use plastics as part of its national sustainability agenda; however, the specific regulatory guidelines for implementing biodegradable and compostable bioplastics remain underdeveloped. This study aims to formulate practical and scalable guidelines for biodegradable and compostable bioplastic products, with a focus on the Federal Territory of Kuala Lumpur as a pilot case. Using a stakeholder-driven approach, a series of focus group discussions (FGDs) was conducted with key representatives from government bodies and the bioplastics industry. The guideline development process encompassed the identification and standardisation of terminology, definition of scope, certification frameworks, regulatory alignment, implementation strategies, and compliance mechanisms. The findings reveal a consensus among stakeholders on the need for clear and harmonised definitions to prevent ambiguity, as well as for certification protocols and enforcement mechanisms to align with existing legal frameworks. Revisions were proposed to terms, scope, and timelines to ensure legal compatibility and practical enforceability. The proposed guideline framework offers substantial potential for national adoption, contingent on inclusive stakeholder engagement across all Malaysian states to ensure uniformity and contextual relevance in its implementation. This study advances Malaysia’s SDG commitments by promoting sustainable bioplastics guidelines, encouraging national adoption through stakeholder engagement, and emphasising future integration of the life cycle assessment (LCA) to enhance the policy’s impact.

## 1. Introduction

Bioplastics consist of different types of plastic, which are characterised by their properties. There are bio-based bioplastics, which are produced from renewable resources such as corn, sugars, and potatoes, and are made by a range of microorganisms [1]. Additionally, there are fossil-based bioplastics, which are derived from petroleum. The degradation process of bio-based bioplastics also varies. Some bio-based bioplastics are biodegradable, while some are non-biodegradable. Certain bioplastics degrade within a defined duration and environment and can be referred to as compostable bioplastics, regardless of their raw material [2]. The term ‘biodegradable’ refers to materials that can disintegrate or break down naturally into biogases and biomass, mostly carbon dioxide and water, as a result of being exposed to a microbial environment and humidity, such as that found in soil [3]—thereby reducing plastic waste. There are several significant advantages of using biodegradable bioplastics. These include a potentially lower carbon footprint, lower energy costs in manufacturing, the avoidance of the use of (scarce) crude oil, reductions in long-lived litter, and improved recyclability [4]. Compostable bioplastics, in contrast to other forms of plastic, decompose into CO_2_, water, inorganic compounds, and biomass within a specific timeframe and under particular conditions. While the terminology and types of bioplastics are well-established, there is a limited empirical assessment in Malaysia regarding their degradation kinetics, carbon footprint, and performance under local waste management conditions. These material-specific data are crucial, as regulatory frameworks must align with actual polymer behaviour and life cycle impacts.

In some countries, various strategies have been employed to address problems related to the production and use of bioplastics, including the implementation of policies and regulations. Policies have the power to shift companies’ focus to different approaches, either locally or as international protocols. Nevertheless, despite the establishment of the United Nations’ 17 Sustainable Development Goals (SDGs), the transition toward environmentally friendly plastics remains slow. Therefore, there is a need for country-specific policies. For example, in Thailand, the government has recognised the bioplastics industry as a strategic industrial sector since 2006, leading to the establishment of the National Roadmap for Bioplastic Development by the National Innovation Agency in 2008 [5]. This policy focused on four components, including sufficient availability of biomass feedstock and raw materials to produce bioplastic products. To broaden knowledge of bioplastics and support the supply chain, the Korean Bioplastics Association (KBPA) has established an authentication scheme for biomass materials, creating a system used to identify bioplastics products. The designation will only be awarded when more than 25% of a substance is composed of ingredients extracted from biomass [6]. At the same time, the European Commission (EC) has issued strategies and directives to mobilise member states and promote a transition to a greener and prosperous economy, which includes the production and use of bioplastics. To that end, the Circular Economic Action Plan was implemented in 2015, with a revision made in 2020 [7]. The Action Plan focused on reducing the use of cheap plastic carrier bags, as it identified packaging materials, especially plastic packaging, as a significant source of waste and an environmental burden. The guideline enables European national governments to reduce their countries’ use of lightweight plastic packaging sacks, thereby minimising the harmful environmental impacts of packaging and packaging waste. The guideline also seeks to increase manufacturers’ liability, in addition to the existing prohibitions on the use of drinking straws, stirrers, or cups. However, most international frameworks integrate life cycle assessment (LCA) and carbon accounting to evaluate the net benefits of bioplastics, while Malaysia’s current roadmap lacks such data-driven evaluation. Without understanding the material science and environmental trade-offs, local adoption risks being symbolic rather than impactful.

However, no specific policy initiatives exist within the European Commission to promote the bioplastics sector. Member states are free to choose appropriate steps of their own to meet these objectives. The National Plastics Plan was launched in Australia in 2021, with the aim of addressing the growing plastics crisis in the country [8]. At the same time, Australia also provides guidelines on how and where to dispose of different types of plastics, aiming to educate and increase awareness among the public, thereby reducing the littering problem. This is because not all biodegradable bioplastics are compostable; some may take years to biodegrade, and some may still leave behind residues [9]. Unlike biodegradable bioplastics, all compostable bioplastics are biodegradable and produce humus, which is beneficial for fertilising and improving soil health. This article focuses on biodegradable and compostable bioplastics. Table 1 presents a summary of international strategies and regulatory approaches to bioplastics.

According to Moshood et al. [10], there are few regulations on bio-based plastic worldwide, just as there are few regulations on bioplastics and other materials. This is in contrast to biofuels, for which policy reviews are underway in various countries. Malaysia is one of the countries where the use of bioplastics remains largely unregulated, with no standardised procedures in place for their production, application, or disposal. This regulatory vacuum presents challenges for policymakers, industry stakeholders, and enforcement bodies alike. Despite the launching of government initiatives to replace fossil-based plastics with bioplastics through Malaysia’s Roadmap Towards Zero Single-Use Plastics, there has been no national or standardised guideline on the production, use, disposal, or treatment of bioplastics in the country. The roadmap exclusively addresses the issue of single-use plastics by promoting the adoption of bioplastics as a green alternative. Hence, this study is crucial, as it aims to develop national guidelines on plastics in general, and bioplastics in particular. The specific objective is to establish guidelines for biodegradable and compostable bioplastics in one of the prominent areas in Malaysia, the Federal Territory of Kuala Lumpur, as a precursor to the national guidelines. The scope of bioplastics in these guidelines refers only to bio-based, biodegradable, and compostable bioplastics, which will be described in this paper as biodegradable and compostable bioplastics. Fossil-based bioplastics are not included in the scope of these guidelines.

This study comprised two stages: (1) to identify the issues and readiness of stakeholders in the transition from conventional plastic to biodegradable and compostable bioplastics; and (2) to develop guidelines on the production, use, and disposal of biodegradable and compostable bioplastics in the Federal Territory of Kuala Lumpur, Malaysia. A qualitative approach, consisting of focus group discussions (FGDs), was adopted in both stages, as the subject matter needed to be explored in depth and discussed with panels that knew the subject well. The first-stage FGDs were conducted on 28 April 2020, and aimed to explore perceptions, views, and opinions regarding the transition from conventional plastic to biodegradable and compostable bioplastics. Thematic analysis was employed to analyse the data, and the findings were published in 2021 [11]. In the second stage, FGDs were conducted on 10th November 2020 to discuss the draft guidelines. This article discusses the findings from the FGDs in the second stage and highlights essential elements of the guidelines. The flow of this study is presented in Figure 1, and the discussion in this article will follow the sections of the draft guidelines. This study is novel in that it is the first in Malaysia to assess stakeholders’ perceptions and readiness regarding the transition from fossil-based bioplastics to bio-based biodegradable and compostable bioplastics (Stage 1), and provides critical insights for guideline formulation (Stage 2). In Stage 1, four groups of leading players in the plastic industry were involved: the ministry and government agencies (n = 14); the state government and local authorities (n = 7); technology providers, suppliers, and traders (n = 13), and the ecosystem enablers (n = 7) [10]. The significant findings were incorporated into the draft guidelines discussed in Stage 2. The participants in each stage were carefully selected to ensure the reliability and validity of their feedback and scientific findings. This study is original in that it has articulated the issues regarding transitioning from conventional plastics to bioplastics within the Malaysian waste management framework.

## 2. Materials and Methods

Using focus group discussions (FGDs), this stage of the study explored and highlighted the elements to be included in the guidelines regarding the production, use, and disposal or treatment of biodegradable and compostable bioplastics from the perspectives of the main stakeholders. FGDs are an appropriate method to use because the subject matter has to be discussed in detail with every stakeholder who is directly involved. The participants were carefully selected to establish the reliability and validity of their input [12]. Because time is of the essence for all these stakeholders, the need to gather them together in one room at the same time for an open discussion on issues of concern was carefully considered.

Regarding bioplastics, it was imperative. This permitted the exchange of views and opinions regarding the research subject. Furthermore, ongoing discussions among stakeholders may stimulate creative ideas and thinking, and thus, this study provided an opportunity to gain in-depth insights into participants’ ideas, values, wishes, and concerns.

The protocol employed in the FGD was in the form of draft guidelines developed based on the Kuala Lumpur City Hall (DBKL) standard operating procedure (SOP) on biodegradable and compostable bioplastics, relevant laws and Acts, and inputs from the Stage 1 FGDs. Diverse sources of input were used to enhance the protocol’s inclusiveness and consistency with existing practices.

### 2.1. Participant Recruitment

A total of nine respondents, consisting of three females and six males, participated in the FGDs (Table 2). To ensure that the responses were not biased toward a particular organisation, we included stakeholders from both the public and private sectors. The stakeholders were categorised into four main groups: stakeholders representing government agencies or the state government (P1, P2, and P3); stakeholders representing the manufacturers (P4, P6, and P7); stakeholders from the certification agency (P5); and stakeholders representing manufacturer and retailer associations (P8 and P9). Participants were recruited using purposive sampling. All participants are decision-makers for the organisations they represent, and their organisations are the primary players in the bioplastic industry. The stakeholders from the government were selected because these individuals act as mandate holders, formulators, implementers, and enforcers of regulations and laws related to bioplastics. The stakeholders from the manufacturing industries were selected because they are involved in the manufacturing, distribution, supplying, and selling of plastic or bioplastic. Representatives from the certification body were chosen because every bioplastic product must undergo inspection and testing before being marketed. The FGD was moderated by a task force member involved in the development of Malaysia’s Roadmap Towards Zero Single-Use Plastics initiative (2018). The moderator was therefore well-versed in the development and use of conventional plastics and bioplastics in Malaysia. This rendered the moderator reliable and credible enough to handle the discussion.

### 2.2. Data Collection

Participants were contacted via email or phone call before they agreed to participate. Before the FGDs, the participants provided informed consent and provided demographic information. The draft guidelines were distributed to the participants before the FGD sessions. This was to enable them to review the policies and familiarise themselves with the issues that would be discussed during the FGD and then consider the draft guidelines within their organisation before the FGD. The FGDs took place at the office of the Malaysian Bioeconomy Development Corporation, and each lasted for 4 h. The FGDs were conducted on 10th November 2020. With the participants’ permission and to preserve their anonymity, audio tape recorders were used in conjunction with handwritten notes. In this study, data collection was triangulated by involving multiple stakeholders from every dimension of the bioplastic industry.

### 2.3. Data Analysis

The FGDs were recorded digitally, and the data were transcribed verbatim. At this stage, a comparative analysis was conducted on the data, as the discussions focused on the same subject [13], namely, the sections of the draft guidelines. Every statement was carefully compared and assessed, and the guidelines were modified according to the results of the comparison and assessment. The identities of the participants were replaced with codenames to maintain their anonymity. To ensure the trustworthiness of the analysis [14], the findings were sent to the participants to verify whether the interpretations accurately reflected their actual meanings, definitions, and perceptions.

## 3. Results

Overall, the guidelines were agreed upon by the participants, although some changes were made to a few sections or items (Table 3). All the participants shared valuable views and opinions on how to improve the suggested guidelines. At the same time, they applauded the move to develop national guidelines on the production, use, and disposal of biodegradable and compostable bioplastics, as this will help stakeholders across the nation understand their responsibilities and recognise their roles in the initiative. Additionally, educational programs can be developed to increase awareness and knowledge of the benefits and risks associated with biodegradable and compostable bioplastics.

The participants involved in the FGD were representatives from the federal ministry, local authorities, manufacturers of bioplastic products, certification boards, and both retailers and national plastic associations. They highlighted several issues in every section, primarily related to the definitions of biodegradable and compostable bioplastics, the consistency of terms used in the guidelines, the scope of the guidelines, the principles and concepts of implementation, the sources of authority, the exceptions, and the tools and mechanisms of compliance, enforcement, and monitoring. The revised draft was returned to the participants for validation.

### 3.1. Introduction

As to the Introduction section, a panel member highlighted that plastic bags, despite their small size and lightweight nature, often account for thousands of pieces within the total waste collected. P8 challenged this:

“6200 plastic bags are small portions since they are light and small. This statement will give the impression that plastic bags are not the problem, but littering is. You may need to find a new reference for this. Again, in the section part of the introduction, it gives us the impression that rubbish comprises only single-use plastic, whereas there are other things too” [P8]

P5 also challenged the claim regarding the impact of single-use plastic on health.

“There is no health threat data. Does this mean it is not a threat to human health, indirectly or indirectly?” [P5]

According to P5, this claim should be followed by evidence or statistics to enhance users’ understanding.

The plan for the guidelines is to eliminate the use of conventional plastic bag carriers and replace them with biodegradable and compostable bioplastic bag carriers. However, there was an argument over the use of the term ‘alternative’ in the Introduction section.

“The term ‘alternative’ needs to be replaced with ‘replacing’ because this statement gives the impression that the use of conventional plastics is still permitted. Hence, it is better to use the term ‘replacing’.” [P5]

The discussion in the Introduction section also calls for attention to be paid to other perspectives.

P8 argued:

“I think it is better to just stick to conventional plastic carrier bags rather than replacing them with biodegradable bioplastics. As I mentioned earlier, the problem is not the plastic itself, whether conventional or biodegradable, but rather the issue of littering. A biodegradable bag is certified as compostable or degradable in a controlled laboratory or industrial composter for a period of 6 months with the presence of microbes. If it is littered or discarded anywhere except at the composting or recycling facilities, this biodegradable bioplastic will not biodegrade. It is better to use conventional plastic as it has a lower carbon footprint, and it is recyclable.” [P8]

P8 argued that the biodegradable properties of the plastic only materialised in a controlled experimental environment and not in a natural environment. Therefore, if littering can be controlled, conventional plastic is more environmentally friendly in terms of its carbon footprint and recyclability potential.

“If we want to reduce the impact of plastic on the environment, in my opinion, we should just prohibit the use of plastic, regardless of whether it is bio or conventional.” [P9]

Conversely, P9 suggested that all plastic, whether biodegradable or conventional, should be prohibited, as both types harm the environment.

“I agree that we should stop the distribution and use of conventional plastic bags and replace them with biodegradable bioplastics, but I think we need a baseline study on the environmental pollution caused by both types of plastic and how much the current and future waste disposal costs are that are required to dispose of conventional and biodegradable bioplastics.” [P1]

P1 somewhat agreed with the replacement of conventional plastic bags with biodegradable plastics but insisted that a baseline study be conducted to identify the actual degree of pollution resulting from both types of plastic, and a costing of waste management for each. According to P1, such data is important for determining the appropriateness of the initiatives and the correct measures to be taken.

### 3.2. Purpose

Using the correct term when developing manual guidelines is extremely important because different terms can carry different meanings, and when these words are used in a sentence, this can significantly alter the overall meaning. This was highlighted by most stakeholders, who believed it should not be taken lightly.

P1 raised some concerns over the use of the term ‘plastic bag’ in the draft guidelines. P1 argued that the bags under discussion should be differentiated from conventional plastic bags. The term should not be used to refer to both types of plastic bags, as users of these guidelines will become confused about which type of plastic bag it relates to.

“We already have this Malaysia Road Map to biodegradable bioplastic bags, and in that document, the term ‘bio bag’ is referring to a plastic bag made from biomaterials. This term is applied and used consistently in the whole document.” [P1]

The interchangeable use of two different terms, ‘biodegradable’ and ‘biodegradable and compostable’, should be avoided.

“The term used, either biodegradable or biodegradable and compostable, needs to be consistent in the whole document. I found that in some sections you use only ‘biodegradable’ when referring to bioplastic products, but sometimes you use the term ‘biodegradable and compostable’. It will be better if all the terms for bioplastic or how you refer to bioplastic products in this guideline are the same.” [P1]

### 3.3. Scope

The draft guidelines include two scopes: the scope of the area and the scope of the business premises. The scope of the area covered in the guideline is the Federal Territory of Kuala Lumpur, considered as a pilot project before implementation at the national level. One participant emphasised the need to engage with every local authority to capture their views and opinions, if these guidelines are to be enforced at the national level.

“The discussion in the future should involve other local authorities before we finalise everything. This is because we need to centralise and coordinate the guidelines so everyone’s opinions are involved, and it will cover other important issues highlighted by other authorities from other states. Engagement and buying sessions with every state in the country should be adequately conducted.” [P7]

About the scope of business premises, it was agreed that the use of biodegradable and compostable bioplastics will be made compulsory at business premises, including privately owned stalls, markets, hypermarkets, big supermarkets, grocery stores, sundry shops, fast food outlets, pharmacies, franchise stores, industries, food outlets, and the convenience shops at all gas stations. However, one participant disagreed with categorising food outlets and fast-food outlets as two separate business premises.

“I think a fast-food outlet is also a food outlet, so it should be together. If we separate them, some might question, How about other food outlets? Why is it not included? Therefore, I suggest that we remove the fast-food outlet from the list and use the food outlet instead. Then it will include a fast-food outlet too.” [P3]

All participants agreed with P3’s suggestion that the list of business premises should be revised.

### 3.4. Objectives

All participants agreed on the proposed objectives of the draft guidelines.

### 3.5. Background

During the discussion on the statement in the Background section, all participants agreed that the statement provides users with a brief amount of information on what has been achieved in the Federal Territory regarding the prohibition of conventional single-use plastics and the implementation of biodegradable bioplastic usage in the Federal Territory of Kuala Lumpur. However, a minor adjustment was suggested by one participant:

“I suggest that we replace the word ‘banned’ with ‘prohibited’. And double-check the date of the meeting with the cabinet association of the Ministry of Federal Territory, as I think it is incorrect, and in the second part of the background, the date of endorsement on the use of biodegradable bioplastic within the federal territory was not stated. And what are the decisions of the endorsement? It is not mentioned here. I suggest that we briefly mention the results from that discussion here for our knowledge and the guideline user too.” [P2]

The concerns of P2 as to the dates associated with the endorsement are justified, as timing is crucial, whether before or after the rules or regulations are enforced. There were no other comments from the remaining participants on the Background section.

### 3.6. Principles and Concepts of Implementation

The Federal Territory has banned the use of single-use plastic bag carriers since 2017; thus, all business premises are prohibited from distributing non-biodegradable single-use plastic bag carriers to their customers. Business premises within the vicinity of the Federal Territory must use biodegradable and compostable bioplastic bag carriers. However, P2 raised some concerns regarding the date mentioned in the statement:

“It was written that the ban on the use of conventional single-use plastic bag carriers in the Federal Territories came into effect on the 1st of January 2017. However, I believe the enforcement date is inaccurate. Please kindly refer to the bio product implementation timeline in the Federal Territory of Kuala Lumpur.” [P2]

“And again, can you please check the date when the RM0.20 charge for plastic bag carriers was implemented, as I think the date on this guideline is incorrect. You can refer to the press conference given by the Federal Territory Minister on the KWP website. If I am not mistaken, it was on the 12th of March 2019 and not the 15th of March 2019.” [P2]

Again, correct dates were emphasised by P2 as timelines are of prime importance in the exercise of enforcement or judicial proceedings. Apart from the accuracy of dates, one participant also stressed the importance of accurately certifying bioplastics to prevent the duplication or counterfeiting of products.

“And do mention in the statement the name of the certification to show what kind of certified bioplastic is allowed to be used on business premises. This is to prevent duplication and the distribution of fake bioplastics on the premises. Only biodegradable bioplastic with a certified eco label from SIRIM is allowed based on SIRIM ECO 001: 2018.” [P5]

P5’s concern is logical as duplicated and fake products could significantly compromise the overall quality of the biodegradable and compostable bioplastics and would eventually thwart the enforcement process.

As of 2019, all business premises in the Klang Valley have been permitted to charge MYR 0.20 (RM) for each plastic bag carrier provided to their customers. However, there was some argument regarding the term ‘allow’ as several participants thought that the correct term should be ‘compulsory’.

“I suggest that we change ‘business premises are allowed to charge RM0.20 for each plastic bag carrier’ to ‘all business premises must charge RM0.20 for each plastic bag carrier’. This statement sounds firmer.” [P9]

P9’s argument is logical, as the term ‘allow’ would leave ample room for excuses.

### 3.7. Definition

In terms of definition, all participants agreed that the definition needs to be carefully explained and consistent with other established documents about biodegradable bioplastic products. In the draft guidelines, a plastic bag carrier is defined as a single-use plastic bag carrier (non-biodegradable or biodegradable and compostable). However, for P7, a plastic bag carrier is not single-use, whether it is biodegradable or non-biodegradable.

“I disagree with the term ‘use of single-use plastic bag carrier’ in the definition because a plastic bag is not single-use. It can be recycled, be it physically, chemically, or mechanically. Maybe we can remove the term ‘single-use’ and use ‘plastic bag carrier (biodegradable or non-biodegradable).” [P7]

Regarding the definition of single-use biodegradable and compostable plastic bags, it was suggested that this definition should be consistent and aligned with the standard operating procedure (SOP) for biodegradable products in the Federal Territory of Kuala Lumpur.

“We can also refer to SIRIM for the SOP or standard definition of each of these items so that it will be consistent between the guidelines, SIRIM, and KWP SOP of biodegradable products.” [P7]

### 3.8. Source of Authority

All participants agreed on the suggested Act (7) and Undang-Undang Kecil, UUK (5), which are listed in the draft guidelines. However, one participant highlighted that this law and the Act need to be consistent with the UUK used by Kuala Lumpur City Hall (DBKL).

“We should refer to the law and the Act in UUK DBKL for this list to avoid redundancy.” [P2]

“Yes. We should refer to UKK DBKL, as I am not so familiar with all the Acts and laws listed in the draft.” [P7]

According to P9, there is a need to introduce a national policy to ban the use of conventional plastic bags altogether. During the FGDs, P9 mentioned that instead of alternating conventional plastic with bioplastics, a better option would be to ban the use of traditional plastic bag carriers.

“The use of conventional plastic bags should be banned completely. I think there is a need to introduce legislation for conventional plastic bags at a national level.” [P9]

Addressing another aspect, one participant suggested that the local authority or enforcement officer should be given the right to confiscate items that the organisations or companies claim are bioplastics but are not.

“In the draft guidelines, there is a mention of various actions taken by the local enforcement authorities when manufacturers or retailers do not comply with the regulations. However, it fails to mention who has the authority to confiscate items that do not comply with the regulations. It needs to be included as individuals or parties need to know who has such authority, including to confiscate items from them (manufacturer or retailers).” [P3]

### 3.9. Compliance

All participants agreed that all business premises are prohibited from providing their customers with non-biodegradable plastic bag carriers. There are cases where manufacturers have claimed that their products were made from biodegradable materials when they were not. Without proper and standardized labelling on the products certified by an authorized certification body, it will be difficult to determine this. Therefore, it should be compulsory for business premises to use biodegradable bioplastic products certified by an official and authorised certification body. Biodegradable bioplastic products must carry the SIRIM ECO 001: 2008 certification label.

“You need to correct the certificate label. It is written as SIRIM Eco-001:2018. It should be SIRIM ECO 001: 2018. The label needs to be clear, precise, and consistent. All bioplastic bag carriers need to be printed with a logo (KWP), SIRIM eco label, which includes the logo, the term ECO Label, SIRIM, environmental claim (either biodegradable or compostable), the standard code (SIRIM ECO 001: 2018), and license number.” [P5]

Some participants explained that the bioplastic bag carrier should only have the label mentioned above. At the same time, the logos specific to the Ministry of the Federal Territory (KWP) and the different states need not be included in the printing.

“I think there is no need for each plastic bag carrier to be printed with a logo specifically for each state. This is because the products may need to be printed separately according to batches for different states.” [P1]

P1’s suggestion is reasonable as production volume has a significant influence on the production costs. If biodegradable and compostable bioplastics are produced in batches, it may increase the production costs for individual bags.

Additionally, the price for each bioplastic bag carrier needs to be included in the guidelines.

“The charge for each plastic bag carrier should also be included in the guidelines in the monitoring section. They (consumer, producer, and retailer) need to know how much they should charge for each biodegradable and compostable bioplastic bag carrier, which is RM0.20.” [P2]

However, P8 raised concerns that the charge mentioned is too small to cover the cost of producing biodegradable and compostable bioplastic bag carriers. They contended that compostable bags are more expensive, have weak characteristics, and break easily. They also require a controlled or industrial composting condition for the waste to biodegrade.

### 3.10. Exceptions

In the draft guidelines, exceptions to the regulation on the use of biodegradable and compostable bioplastic bag carriers were suggested for a limited number of business premises, including wet markets, pharmacies, and launderettes. These premises might be allowed to use conventional plastic bags. However, P2 and P4 disagreed:

“Consumers should bring their own plastic bag or grocery bag, or shopping basket when purchasing items at the wet market.” [P2&P4]

The participants agreed not to allow any exemptions, in order to prevent users from taking advantage of or misusing the allowance.

### 3.11. Enforcement and Monitoring

For enforcement and monitoring purposes, participants felt the term ‘compound notice’ should be used instead of just ‘notice’. This will give a clear indication of the type and weightage of the notice. There should also be an indication of how many offences are required for the enforcement officer to issue a court notice to the offender.

“Compound notice needs to be used in this section.” [P9 & P4]

“Should also include an explanation on when a warning notice will be issued, and a compound notice for subsequent offences. Then it will be clear for other stakeholders regarding the order of issuing notices, and the duration of time given to the offender to rectify the issue before further action is taken.” [P9]

As to another issue, it was mentioned that one of the steps taken to ensure the authenticity of biodegradable and compostable bioplastic materials is to conduct a rapid or physical test. However, participants raised the important question of who should bear the testing expenses.

“Here, in this guideline, a rapid or physical test on the bioplastic is mentioned to make sure it is made from biodegradable bioplastic materials. But who should pay for the testing? Is it SIRIM, the manufacturer, or the retailer? It is important to include this because at the end of the day, somebody must pay for that.” [P3a & P5]

The concerns raised by P3a and P5 are valid, as the party that must bear the cost of testing needs to be specified in the guidelines for that party to take responsibility.

“I think in terms of the workflow chart, we can refer to the Standard Operating Procedure for Enforcement of Bio-product in the Federal Territory that the Ministry of Federal Territory has approved.” [P2]

In line with the suggestion to ensure the law and Act in the guidelines are consistent with the existing law and Act, P2 advised that the workflow chart should be consistent with the established SOP on bio product enforcement, to increase user understanding.

### 3.12. Closing

P1 and P4 again highlighted the inconsistencies in the use of the term ‘bioplastic’, which appears in the closing section instead of ‘biodegradable and compostable bioplastic’.

From Section 3.1, Section 3.2, Section 3.3, Section 3.4, Section 3.5, Section 3.6, Section 3.7, Section 3.8, Section 3.9, Section 3.10, Section 3.11 and Section 3.12, stakeholder validation highlighted three recurring themes: (1) the need for harmonised terminology, with eight out of twelve term-related suggestions accepted; (2) requests for alignment with local Acts and SOPs to avoid redundancy (five suggestions, all accepted); and (3) calls for baseline pollution and cost assessments before nationwide adoption (two suggestions, both flagged for future consideration). These patterns reflect the stakeholders’ prioritisation of regulatory clarity and enforcement feasibility over broader environmental modelling, underscoring the necessity for standardisation before scaling the guidelines.

Building on these findings, the stakeholder feedback can be synthesised into three analytical clusters that underpin the refinement of the guidelines. The first cluster, Terminology and Definition Consistency, encompasses issues highlighted in Section 3.1, Section 3.2, Section 3.7 and Section 3.12, in which stakeholders stressed the importance of precise and harmonised terms to avoid ambiguity in compliance and enforcement. The second cluster, Regulatory and Enforcement Alignment, draws on feedback from Section 3.5, Section 3.6, Section 3.8 and Section 3.11, underscoring the need for consistency with existing local laws, such as the UUK DBKL, and the accuracy of enforcement timelines and procedures to ensure operational feasibility. The third cluster, Economic and Operational Considerations, reflects concerns raised in Section 3.6, Section 3.9 and Section 3.10 regarding the implications of cost structures, including the MYR 0.20 pollution charge, certification labelling, and the practical adjustments required by manufacturers and retailers. These clusters collectively form the foundation for the subsequent discussion (Section 4), in which the guideline elements are critically assessed within the broader framework of sustainability and compliance.

## 4. Discussion

Using the correct term when developing guidelines that various users or stakeholders will use is extremely important. All stakeholders highlighted the importance of this during the FGD sessions. Developing specific guidelines for a particular industry or field requires the use of correct and precise terms to convey the right information to the target audiences. Using terms that seem ambiguous and vague may lead to varying interpretations of the guidelines by different stakeholders. This can become a loophole. According to the stakeholders involved in the FGD sessions, several terms in the drafted guideline need to be changed, for instance, using ‘prohibited’ instead of ‘banned’ and ‘replacing’ instead of ‘alternative’, and ‘plastic bag’ being replaced by the specific type of plastic bag. Although these may not create a huge misunderstanding for some, they may have enormous consequences for certain aspects of the bioplastic industry. This is because ambiguity and vagueness in the use of terms can reduce the likelihood of adherence to the guidelines [15]. According to both authors of the cited study, ambiguity leads to inconsistent interpretation and, in turn, to inappropriate practices and errors. Ambiguous terms can be interpreted in more than one way, and vagueness occurs when the boundaries of a word’s meaning are not well defined.

The terms ‘biodegradable bioplastics’ and ‘biodegradable and compostable bioplastics’ are also quite problematic. There are various types of bioplastics with distinct chemical, physical, and mechanical properties, as well as different methods for disposal and environmental impacts. Such concerns have also been highlighted in other studies, where the term ‘bioplastic’ is often used loosely and synonymously with ‘biodegradable,’ although not all bioplastics are biodegradable. In a more specific sense, a biodegradable plastic is a plastic material that complies with specific official standards of biodegradability, which require a certain amount of degradation to be scientifically observed within a specified time frame and under particular conditions [16]. Similarly, compostable bioplastic needs to undergo biodegradation in industrial composting facilities and must comply with specific standards. Therefore, using both ‘biodegradable bioplastics’ and ‘biodegradable and compostable bioplastics’ in the same documentation can be highly confusing to stakeholders. These guidelines, however, will consistently use the term ‘biodegradable and compostable bioplastics’ to be aligned with the existing Roadmap Towards Zero Single-Use Plastics.

Another concern highlighted during the FGDs was the enforcement date and certification label. This is because both were noticeably incorrect in the draft guidelines. The dates on which the use of bioplastics and the imposition of charges on non-bioplastic carrier bags were enforced at business premises need to be accurate. This concern is crucial because the permission to issue the notice and penalty, as well as the basis of investigation and prosecution, depend crucially on the date of enforcement. Incorrect enforcement dates can render the manufacturer susceptible to compound notices from the authority and may also result in erroneous investigations and prosecutions.

The ‘scope’ of the guidelines refers to the area or coverage where a specific guideline is implemented for information and monitoring, as well as the type of business premises it covers. This guideline was developed for the Federal Territory of Kuala Lumpur, under the jurisdiction of Kuala Lumpur City Hall, also known as DBKL. Currently, the implementation of biodegradable and compostable bioplastics products policies varies among local enforcement authorities. These variations have resulted in overly complex guidelines, making these guidelines difficult to adhere to. Therefore, it is recommended that national guidelines be developed to provide uniform instruction and regulation. Before finalising them, engagement and buy-in sessions with stakeholders from other states or local authorities need to be conducted to reach a consensus. Broadening the scope of the guidelines to the whole nation can save manufacturers and enforcement personnel a substantial amount of time and costs, as they only need to refer to one set of similar guidelines. To ensure the availability of appropriate resources and efficient enforcement, effective communication and alignment on the guidelines are necessary among stakeholders. This will also ensure that consumers receive correct and consistent information regarding the implementation of the guidelines. Furthermore, information on the scope of business premises is also essential, as it alerts those who manage the premises to their responsibilities.

Definitions are the most crucial section in every legislative text, as they control the meaning of terms [17]. They should therefore be precise and indisputable. In the proposed guidelines, the definitions of single-use non-biodegradable plastic carrier bags and single-use biodegradable and compostable plastic carrier bags are problematic, as the term ‘single-use’ can be disputed. According to the stakeholders, a plastic bag can be used multiple times before it wears out, so the term seems inappropriate. The term ‘single-use’ plastic bag carrier is defined broadly, as encompassing all carrier bags supplied to be used only once, to carry goods away from a point of sale. However, a single-use plastic bag can be converted into a reusable plastic bag carrier and used multiple times, resulting in less serious environmental impacts than those initially claimed [18]. In other words, if the non-biodegradable plastic is used numerous times, it will not fall into the category of plastic defined in these guidelines. The stakeholder suggested omitting the term ‘single-use’ from the definition and retaining only the characteristics of being either non-biodegradable, biodegradable, or compostable.

The use of a law and an Act to support the implementation of the guidelines is deemed necessary to provide enforcement officers with the essential terms of reference to implement them. According to the stakeholders in the FGD sessions, the law and the Act referenced in the guidelines must be consistent with those used in the Undang-undang kecil (UUK) by the Kuala Lumpur City Hall (DBKL). The current UUK by DBKL has been successfully amended to accommodate the required authorisation by the local authority and should become the primary reference for local authorities in other states in Malaysia. Local authorities or enforcement officers should be given the authority to confiscate items that do not comply with the required standards. This concurs with the suggestion by Glanowski [19] that governments should pay attention to weak enforcement and give local enforcement officers the legal power to prosecute offenders as well as the right to confiscate non-compliant items. Therefore, there should be better provisions in law regarding the authority of enforcement officers, to enable them to do their jobs effectively.

There was a suggestion by stakeholders that a law should be enacted to ban the use of conventional plastic bag carriers completely. Some opined that having such a law could help to accelerate the shift from traditional plastic bags to biodegradable and compostable plastic bag carriers. However, according to Asmadianto, Arfah, and Krismiyati [20], the ban on the use of plastic bags currently in effect in several parts of the world has been ineffective because environmentally friendly shopping bags are not affordable. Before implementing a ban on plastic bags, a plastic substitute should be available on the market at an affordable price. Furthermore, a complete ban on conventional plastic production may have a snowball effect on industries related to the plastic industry, such as textiles or the food industry [21].

A certificate is an official document used to guarantee specific characteristics of products [22]; in this case, it distinguishes bioplastics from conventional plastics. According to the stakeholders in this study, each biodegradable or compostable bioplastic carrier bag should be printed with an eco-label logo from the certification body, and retailers should be allowed to charge MYR 0.20 per carrier bag. Having an eco-label or logo on the carrier bag that indicates it is either biodegradable or compostable will inform consumers about the type of plastic bag they are receiving from the business. According to Chekima and Borin [23], eco-labels or certifications are valuable to consumers because they build trust in products, communicate specific product properties, and promote the purchase of environmentally friendly products. This concurs with a previous study in Germany by Emberger, Klein, and Menrad [24], which revealed that product certification was given the second-highest average importance, underscoring the importance of labelling and information for many consumers. Certification of bioplastics is important, as it offers consumers a choice and provides information about the correct handling of the product after use [25]. In addition, the price quoted by the stakeholders is very minimal, and affordable for consumers. According to Sherer et al. [25,26], prices are often a key factor in determining consumer choices, and most respondents preferred low prices for bio-based products. However, it is argued that manufacturers only need to comply with having the eco-label on their product. The notion that these products would also need to print a specific logo representing each state elicited negative feedback from some stakeholders during the FGDs. Standardising eco-labels for all biodegradable bioplastic products is both cost- and time-effective for manufacturers as well as for enforcement officers conducting field monitoring. In line with reporting the correct dates, producing a correctly labelled product is also essential, as manufacturers of biodegradable bioplastic bag carriers can easily make errors in printing the certificate label on their products if the details are incorrectly described in the guidelines, which may incur additional costs and administrative burdens.

The implementation of biodegradable and compostable bioplastic guidelines requires strong and consistent enforcement, as well as effective monitoring, to ensure that manufacturers and business owners comply. Local enforcement officers must be familiar with the laws, Acts, regulations, guidelines, and policies governing the use of bioplastic products. It is argued that they need to be knowledgeable to conduct their job effectively and efficiently, ensuring that parties involved in the manufacturing and use of bioplastics comply with the guidelines. Therefore, a clear and comprehensive flowchart is vital for any set of guidelines. According to the stakeholders in this study, the draft guidelines’ flowchart lacks some vital information, for example, the designation of who will bear the testing cost when a local enforcement officer sends a ‘suspicious’ item to test whether manufacturers are complying with the guidelines. Additionally, for follow-up purposes, a time frame for each compound notice is required in order to inform both the enforcer and the offender. An accurate and detailed flowchart is essential, as it serves as a reference point for all stakeholders in the bioplastic industry.

By engaging key stakeholders from government, industry, and certification bodies, this study emphasises the need for a structured implementation that not only supports regulatory compliance but also aligns with broader sustainability goals. The proposed guidelines contribute directly to several Sustainable Development Goals (SDGs), including responsible consumption and production (SDG 12), climate action (SDG 13), and life below water (SDG 14), through improved waste management and reduced reliance on fossil-based plastics. Although the guidelines do not yet incorporate a complete life cycle assessment (LCA), the study acknowledges the importance of the LCA in evaluating the environmental impacts of bioplastics throughout their entire life cycle, including the production, use, and disposal phases. A preliminary LCA comparison, based on published emission factors [16,24], suggests that replacing 50% of Kuala Lumpur’s annual single-use plastic consumption with certified compostable bioplastics could yield a net GHG reduction of 4500–5800 tonnes CO_2_-eq per year. Although bioplastic production has higher upfront energy intensity, end-of-life compostability offsets landfill methane emissions and reduces persistent plastic leakage. Integrating such LCA metrics into the guideline framework would align Malaysian policy with global standards and provide evidence-based justifications for stakeholders and regulators. Future policy development should integrate LCA tools to support informed decisions and promote environmentally responsible alternatives to conventional plastics.

In brief, this study highlighted the strengths, weaknesses, opportunities, and threats (SWOT) associated with the transition to and implementation of bioplastics guidelines in Malaysia. The initiatives have received strong support from policymakers and regulators in Malaysia, which will significantly strengthen the process. This strength, however, is weakened by the absence of the appropriate infrastructure and mechanisms vital to the process. The production of bioplastics on a large scale is expected to promote growth in the green sector, resulting in job opportunities and new market opportunities. However, resistance from major stakeholders is an anticipated threat, as this shift has significant cost implications.

## 5. Conclusions

This study examined stakeholder feedback on draft guidelines for biodegradable and compostable bioplastics in the Federal Territory of Kuala Lumpur, aiming to utilise this local initiative as a foundation for future national guidelines. The feedback showed an explicit agreement on the importance of having clear, standardised, and practical guidelines. Given the growing interest in replacing conventional plastics with more sustainable alternatives, especially within the framework of a circular economy, these guidelines play a key role in ensuring that bioplastics are produced, used, and disposed of responsibly. This study also clearly demonstrated how these guidelines support SDGs 12, 13, and 14.

Stakeholders highlighted several key areas that need attention: consistent and well-defined terms, alignment with existing laws and policies, clear procedures for certification and enforcement, and realistic timelines. Suggestions were also made to refine specific sections of the draft, particularly those on the scope, classification of materials, and the authority of enforcement officers. The study also pointed out the need to involve stakeholders from other states through engagement sessions to ensure broader acceptance and practical implementation across Malaysia. Simplifying compliance requirements and improving communication about the guidelines were also seen as necessary to support industry adoption.

One of the main limitations of this study was the restricted access to minutes from high-level ministerial meetings, as these documents are confidential and not publicly available. Future studies can address this by involving ministry-level representatives to provide more policy insights.

This study offers practical insights for improving policy on bioplastics and contributes to the broader agenda of developing a sustainable circular economy. By introducing structured and widely accepted guidelines, Malaysia can manage biodegradable and compostable bioplastics more effectively, reducing environmental harm while supporting more sustainable production and waste practices. If applied nationwide, these efforts have the potential to shift the plastics value chain towards a circular model that prioritises long-term environmental and social benefits. The findings of this study may seem general. Still, they accurately reflect the generic characteristics of policy-related documents, which are highly useful in developing a realistic and agreeable policy in the future. Without a policy as such, green behaviours, including the adoption of bioplastics, would remain voluntary and unregulated, and impactful breakthroughs in bioplastics studies may never leave the lab.

## Figures and Tables

**Figure 1 polymers-17-02165-f001:**
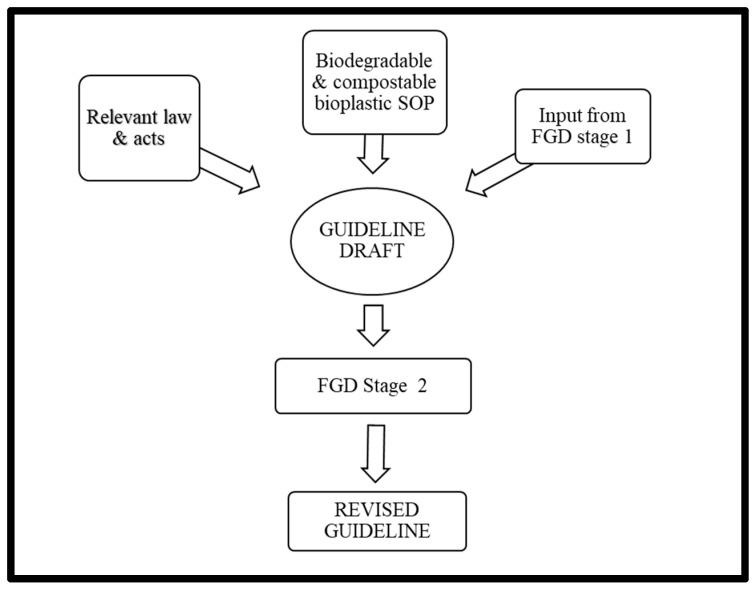
Flow of the study.

**Table 1 polymers-17-02165-t001:** An international summary of strategies and regulatory approaches associated with bioplastics.

Country/Region	Policy or Initiative	Key Features	Certification/Standard	Stakeholder Engagement	Life Cycle Assessment (LCA)	Relevance to Malaysia
European Union	Circular Economy Action Plan (2015, revised 2020)	Reducing plastic packaging, extended producer responsibility, and guidance on biodegradable packaging	EN 13432 (Compostability); national-level certifications	Medium: Driven by EU directives; member states customise	Often integrated in policy evaluation	Emphasises policy alignment and producer accountability
Thailand	National Roadmap for Bioplastic Development (2008)	Focus on biomass feedstock, production support, and industry incentives	National certification schemes (e.g., Thailand Green Label)	High: government–industry collaboration	Not explicitly discussed	Demonstrates government-led strategic planning
South Korea	Bioplastics Authentication Scheme by KBPA	Requires > 25% bio-based content for certification	Korean Bio-labelling System	Medium–High: involvement of industry and R&D	Not specified	Functional model for certification-focused policy
Australia	National Plastics Plan (2021)	Ban on certain plastic types, national education campaign, and guidance on disposal	AS 4736 (Compostable Plastics Standard)	Medium: public awareness focus	Yes—recognised in environmental product evaluations	Highlights the need for public education and clarity on compostability
Malaysia (this study)	Draft Guidelines for Biodegradable and Compostable Bioplastics in FT Kuala Lumpur	Stakeholder-driven guideline development covering scope, enforcement, definitions, and certification	SIRIM ECO 001: 2018	High: Multi-sector FGD with certification bodies, manufacturers, and regulators	Proposed for future integration	A pilot framework toward national adoption and harmonisation

**Table 2 polymers-17-02165-t002:** Participants in the focus group discussion (FGD).

No	Organization	Panellist	Code
1	Ministry of Energy, Science, Technology, Environment & Climate Change (MESTECC)	Secretariat Eco-Innovation Division	P1
2	Ministry of Federal Territory (KWP)	Secretariat Strategic Development Division	P2
3	Kuala Lumpur City Hall (DBKL)	Secretariat Environmental Health Section Department of Health and Environment	P3
4	Malaysian Bioeconomy Development Corporation Sdn. Bhd.	President of the Industry Development Division	P4
5	SIRIM QAS International Sdn. Bhd.	Product Certification & Inspection Department	P5
6	Cardia Bioplastics (Malaysia) Sdn. Bhd.	Secretariat	P6
7	Kaneka Malaysia Sdn. Bhd.	Secretariat	P7
8	Malaysian Plastics Manufacturers Association (MPMA)	Advisor	P8
9	Malaysia Retailers Association (MRA)	Member	P9

**Table 3 polymers-17-02165-t003:** A summary of the proposed draft of the guidelines and the results from the focus group discussion (FGD).

No	Summary of Proposed Draft of Guidelines and Results from FGD	
1	Section 1.0 Introduction	
	Proposed draft to the panel	The excessive production and improper disposal of single-use plastic carrier bags significantly contribute to land, water, and marine pollution, with Malaysia ranking eighth globally in mismanaged plastic waste. Single-use plastic bags commonly end up as litter due to their light weight and mobility; this is compounded by public negligence and weak waste management systems. Studies and clean-up efforts consistently show that plastics, particularly plastic bags, dominate beach and urban waste, worsening environmental degradation. Improperly discarded plastic bags cause direct problems for local authorities, such as clogged drains that contribute to urban flooding, as evidenced in Kuala Lumpur’s 2016 floods. The Ministry of Federal Territories has banned non-biodegradable plastic bags in retail to protect the environment, raise public awareness, cut waste disposal costs, and encourage the use of eco-friendly alternatives.
	Result/Input from FGD	An insignificant amount of single-use plastic in the total collection of waste The absence of health impact data Replace the term ‘alternative’ with ‘replacing’ Conventional plastic is better as it has a lower carbon footprint and is recyclable Prohibit the use of plastic, regardless of whether bio or conventional Conduct a baseline study on the environmental pollution caused by both types of plastics and their waste disposal costs
2	Section 2.0 Purpose	
	Proposed draft to the panel	The Guidelines aim to serve as comprehensive guidance for stakeholders on implementing the use of biodegradable and compostable carrier plastic bags in Kuala Lumpur. They outline definitions, scope, procedures, approaches, and the relevant legal framework to ensure effective and consistent execution of the initiative.
	Result/Input from FGD	The terms plastic bag, conventional plastic bag, biodegradable, and compostable bioplastic bag should be consistent in the whole document and consistent with other established documents
3	Section 3.0 Scope	
	Proposed draft to the panel	The Guidelines apply within the Federal Territory of Kuala Lumpur for enforcing the Use of Biodegradable Products Initiative, focusing on biodegradable and compostable carrier plastic bags. Compliance is required across various business premises, including hawkers, markets, food outlets, trade and industry premises, hypermarkets, supermarkets, convenience stores, fast food outlets, petrol station stores, chain stores, and pharmacies.
	Result/Input from FGD	Engagement with local authorities in every state has to be conducted if the scope of the guideline is to be expanded to the national level Different types of food outlets should be scoped together
4	Section 4.0 Objective	
	Proposed draft to the panel	Provide essential information to ensure the initiatives are implemented effectively and uniformly. Enhance coordination among policymakers, enforcement agencies, and business entities. Secure positive support from consumers and businesses through transparent and inclusive implementation.
	Result/Input from FGD	No amendment
5	Section 5.0 Background	
	Proposed draft to the panel	The ban on single-use non-biodegradable carrier plastic bags and the adoption of biodegradable products in the Federal Territories were decided on 27 April 2016 by the Federal Territories Minister. A notification letter regarding the use of biodegradable and compostable carrier plastic bags was issued to all ministries and agencies on 18 July 2016. A Cabinet Note on implementing biodegradable plastic products in the Federal Territory was presented for Cabinet information on 24 March 2017.
	Result/Input from FGD	Change the word ‘banned’ with ‘prohibited’ Double-check the date of the meeting and endorsement Explain the result of the meeting briefly
6	Section 6.0 Principles and concepts of implementation	
	Proposed draft to the panel	The Federal Territory officially banned single-use non-biodegradable carrier plastic bags effective 1 September 2017, requiring all business premises to switch to biodegradable and compostable alternatives. From 12 March 2019 onwards, business premises may charge MYR 0.20 per single-use biodegradable and compostable carrier plastic bag provided to customers. All such bags must be certified with ECO Label 001 by SIRIM
	Result/Input from FGD	Recheck the date of enforcement in the Federal Territory. Elaborate on the certificate specification Change the word ‘allow’ to ‘compulsory’ to charge MYR 0.20 for each plastic bag
7	Section 7.0 Definition	
	Proposed draft to the panel	Single-use non-biodegradable plastic carrier bags are thin, lightweight, petroleum-based plastics (e.g., LDPE) that cannot be broken down by microorganisms and are typically used for a single purchase. Single-use biodegradable and compostable plastic carrier bags are thin, lightweight plastics made from renewable sources (e.g., starch compounds, biopolymers) that can decompose through microbial enzymatic processes and are also generally used for a single purchase.
	Result/Input from FGD	Omit the term ‘single-use’ Should be consistent with other established documents Include the amount of charge in the pollution charge definition
8	Section 8.0 Source of authority	
	Proposed draft to the panel	Acts, By-Laws and related regulations are: Local Government Act 1976. Licensing of Hawkers (Wilayah Persekutuan Kuala Lumpur) By-Laws 2016. Market Licensing (Wilayah Persekutuan Kuala Lumpur) By-Laws 2016. Licensing of Food Establishment (Wilayah Persekutuan Kuala Lumpur) By-Laws 2016. Licensing of Trades, Business and Industries (Wilayah Persekutuan Kuala Lumpur) By-Laws 2016. Licensing Guidelines, Licensing and Business Development Department, DBKL. Treasury Circular No. 5 of 2009, Government Confiscated Store Management Procedures.
	Result/Input from FGD	Refer to Act (7) and Undang-Undang Kecil, UUK Laws and Acts need to be consistent with local authorities’ laws and Acts Introduce legislation at the national level Elaborate on the level of authority of the enforcement officer
9	Section 9.0 Compliance	
	Proposed draft to panel	Business operators are strictly prohibited from using or supplying non-biodegradable disposable carrier plastic bags; only biodegradable or compostable bags certified with the SIRIM Eco-label (SIRIM ECO-001:2018) are permitted. Operators may charge a minimum of MYR 0.20 for each certified biodegradable or compostable plastic bag provided to customers. Each plastic bag must display the SIRIM Eco-label logo; the words “ECO-Label”, “SIRIM”; an environmental claim stating “Biodegradable and Compostable”; and the standard code “SIRIM ECO 001: 2018”.
	Result/Input from FGD	Products must carry the certification label of SIRIM ECO 001: 2008 The charge for each plastic bag carrier should be included
10	Section 10.0 Exceptions	
	Proposed draft to the panel	Exceptions to the biodegradable products requirement apply to specific plastic products, including: Plastic bags for packing wet or fresh materials (e.g., vegetables, fruits, meat, fish). Plastic bags for laundry or dry cleaning. Plastic bags for prescription drugs. Plastic packaging for live fish.
	Result/Input from FGD	No exemption should be given
11	Section 11.0 Enforcement and monitoring	
	Proposed draft to the panel	Random inspections of business premises will be carried out by Kuala Lumpur City Hall and relevant departments to monitor compliance. A first-offence notice will be issued to any business licensee found using non-compliant products without SIRIM Eco-label certification (SIRIM ECO 001: 2018). If the initial offence notice and compound offer are not settled within the given period, a compound reminder notice will be issued as a final settlement opportunity before court action.
	Result/Input from FGD	The term ‘notice’ changes to ‘compound notice’ Mention the order of issuing notices Specify the party that has to bear the cost of testing The workflow chart should be consistent with other established workflows
12	Section 12.0 Closing	
	Proposed draft to the panel	The Guidelines were developed using field studies, existing reference materials, and consultations with stakeholder representatives. The Federal Territory of Kuala Lumpur retains the right to amend and enhance the Guidelines when necessary. Any amendments must align with the primary purpose of the Use of Biodegradable Products Initiative.
	Result/Input from FGD	Inconsistency of the terms ‘bioplastic’ and ‘biodegradable and compostable bioplastic’

## Data Availability

Data may be available from the corresponding author upon reasonable request, subject to approval by the relevant ethics committee.

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
