# Peer review of "Toward National Guidelines for Biodegradable and Compostable Bioplastics: A Case Study in the Federal Territory of Kuala Lumpur, Malaysia"

_polymers, 2025, doi:10.3390/polym17162165_

Round 1
Reviewer 1 Report
Comments and Suggestions for Authors
Dear authors, many thanks for compiling the article presenting the methodology of developing a guideline for biodegradable plastics in the Federal Territory Kuala Lumpur.
In my opinion, the text is out of the scope of the scientific journal polymers. It does not present scientific findings. Its main part (chapter 3) is the protocol of a 4-hour meeting in which a draft regulation was discussed by some stakeholders. The process described in chapter 2 as well as the conclusions presented in chapter 4 are of a very general nature and do not present any novelty or originality. Finally, the text deals with a meeting that took place five years ago without supplying any information on how the results of the meeting were used in rewriting the draft guidelines!
In case you decide to resubmit the paper to another journal, e.g. one that focuses on environmental policy development, which is more appropriate from my point of view, you might take care of the following detailed suggestions for improving the text:
Lines 32-40: Please delete these writing guidelines and add headline “1. Introduction”
Line 40 and throughout the document: It would be good if you stuck to your own advice of using clear and harmonized definitions. In line 40f, the definition of "Biobased Plastics" is given, not the definition of "Bioplastics".
"Bioplastics" as defined by the European Bioplastics Association and the U. S. Plastics Industry Association "are biobased, biodegradable, or both" (short definition, cf. https://www.european-bioplastics.org/bioplastics/);
Extended definition (cf. https://www.plasticsindustry.org/who-we-serve/recycling-sustainability/bioplastics/):
“Bioplastics are either:
- made from a renewable resource such as corn or sugar cane (biobased).
- break down completely via a natural process (biodegradable).
- both biobased and biodegradable.”
Such clarification of the term "Bioplastics" should be presented at the beginning of the article. Throughout the article, the authors should preferably not use the term "bioplastics", but better specify precisely if they mean compostable plastics, compostable biobased plastics, durable bio-based plastics or whatever combination of the origin (fully or partly biobased, fossil-based) and the end-of-life behavior (compostable, photodegradable, biodegradable etc.) serves to characterize the material under consideration.
The combinations "biodegradable bioplastics" often used by the authors can lead to misunderstandings. If you want to write about biodegradability, use "biodegradable plastics", for plastics made from biomass origin, use “biobased plastics”. If you mean a biobased and biodegradable material, use "biodegradable and biobased plastics". The same applies to the combination “compostable bioplastics”.
Lines 43-44: Delete this sentence (“Biodegradable bioplastics are … visible toxic remainders.”): The message given therein is incorrect and the following sentence (lines 44-47) more precisely and correctly defines biodegradability.
Line 51: It’s “GHG” instead of “GHC”
Line 62: Please clarify that the Thai National Roadmap is about biobased plastics. The same holds true for the Korean Authentication Scheme.
Line 77: Here, the authors introduce the term “biological … bioplastics”. What do they mean, “biobased … bioplastics” or else?
Line 87: Why and how do some biodegradable plastics “leave behind metal residues”? Please explain more clearly or cite a reference from which this statement originates.
Table 1: As can be seen from the entries in the column “Key Features”, the Thai as well as the South Korean Initiatives deal with biobased plastics – and not with “biodegradable and compostable plastics” denoted the table heading!
Lines 96-97: Moshood et al. - in the title of their article - clearly and correctly distinguish between "biodegradable plastic products" and "bio-based plastics". It is highly recommended that the authors of the submitted manuscript also write with such clarity and correctness.
Line 101: Which type of plastics (biobased and/or biodegrdable) does the mentioned Malaysian Roadmap deal with? Or is that Roadmap concerning single-use plastics?
Line 154: At the end of the line, “act” is used as a verb, not a legal provision, and thus written with lower case initial letter.
Lines 191-193: Delete these writing guidelines
Table 3 and chapter 3: The details given in Table 3 and chapter 3 do not present useful information for the readers, unless the underlying basic document - the draft guidelines from summer 2020 - were presented. I suggest making that document accessible to readers as a supplement.
Lines 216-217: Please give the needed reference data for that number. Is that 6,200 plastic bags per person per month, or per household per year, or ...
Line 594: Please correct the typo in Prof. Menrad’s name.
Author Response
Comment 1:
The article is out of scope for Polymers as it lacks scientific findings. Its main part (chapter 3) is the protocol of a 4-hour meeting in which a draft regulation was discussed by stakeholders. The process described in chapter 2 and the conclusions in chapter 4 are very general and do not present novelty or originality. The text discusses a meeting that took place five years ago without indicating how its outcomes were used to rewrite the draft guidelines.
Response 1: We respectfully contend that the manuscript is within the scope of Polymers under the Special Issue: Challenge and Prospect of Plastics and Bioplastics for Sustainable Circular Economy. While the core of the article presents a guideline development process, it directly addresses governance and stakeholder aspects that are essential for implementing biodegradable plastics sustainably. Establishing clear definitions, standards, and local policy frameworks is a necessary complement to scientific advances in bioplastics. By documenting this process, the article supports the Special Issue’s aim to explore practical pathways for advancing sustainable plastic use and circular economy objectives.
Regarding the use of data from the stakeholder discussion and guideline development (GD), we acknowledge that data relevance depends on the stability of the social, cultural, or policy context. Focus Group Discussion (FGD) data remains valid when the underlying context has not changed significantly (O. Nyumba et al., 2018). In our case, the policy landscape for biodegradable plastics in the Federal Territory of Kuala Lumpur remains broadly similar, and the guidelines derived from the original discussions continue to inform current practice.
As noted by Rabiee (2004) and Kuckartz & Rädiker (2019), for stable topics such as the development of local regulatory frameworks and long-standing sustainability practices, older FGD data can retain significant value if properly contextualised, which we have ensured by providing updates and clarifications in our revised manuscript.
Comment 2:
Lines 32–40: Please delete these writing guidelines and add the headline “1. Introduction”.
Response 2:
Line 35 — The authors have inserted the appropriate section heading. (Blue in color)
Comment 3:
Line 40 and throughout the document: Use clear and harmonized definitions. The definition provided is for “Biobased Plastics”, not “Bioplastics”.
Response 3:
Lines 36–55 — The authors have revised the definitions for accuracy and provided clear distinctions of terms throughout the manuscript. Short and extended definitions have been added to ensure consistency.
Comment 4:
Clarify “Bioplastics” using the definitions from credible associations at the beginning of the article.
Response 4:
Lines 36–55 — Please see Response 3 above; the clarification and definitions have been added as suggested.
Comment 5:
Use the term “bioplastics” precisely and specify whether compostable, biobased, or other.
Response 5:
Lines 36–55 — Please see Response 3; the terminology has been revised for clarity throughout the manuscript.
Comment 6:
Avoid combinations like “biodegradable bioplastics” that may cause misunderstandings.
Response 6:
Lines 36–55 — The terminology has been revised to reflect accurate and precise usage throughout the text.
Comment 7:
Lines 43–44: Delete the sentence (“Biodegradable bioplastics are … visible toxic remainders.”).
Response 7:
The authors have removed the sentence as suggested.
Comment 8:
Line 51: “GHC” should be “GHG”.
Response 8:
Line 680-The authors have corrected the abbreviation.
Comment 9:
Line 62: Clarify that the Thai National Roadmap concerns biobased plastics. Same for the Korean Authentication Scheme.
Response 9:
Line 64-The authors have revised for precision, both concern bioplastic.
Comment 10:
Line 77: Clarify the term “biological … bioplastics”.
Response 10:
The authors have revised the wording for precision.
Comment 11:
Line 87: Explain or cite the source for the statement on metal residues.
Response 11:
Line 93-The authors have added a citation to support this statement.
Comment 12:
Table 1: Heading should correctly describe the initiatives as biobased, not biodegradable and compostable.
Response 12:
Line 19-The authors have corrected the table heading accordingly.
Comment 13:
Lines 96–97: Distinguish clearly between biodegradable and biobased plastics.
Response 13:
The authors have aligned terminology for consistency and accuracy.
Comment 14:
Line 101: Clarify the type of plastics addressed by the Malaysian Roadmap.
Response 14:
Line 111-The authors have revised the section to specify the scope accurately.
Comment 15:
Line 154: “Act” used as a verb should not be capitalized.
Response 15:
The authors have corrected the capitalization.
Comment 16:
Lines 191–193: Delete these writing guidelines.
Response 16:
The authors have removed the lines.
Comment 17:
Table 3 and Chapter 3: Make the underlying draft guidelines available as supplementary material.
Response 17:
Line 228-The authors have added the proposed draft in Table 3.
Comment 18:
Lines 216–217: Clarify the reference for the figure “6,200 plastic bags”.
Response 18:
Line 241-The authors have revised the sentences to clarify this figure.
Comment 19:
Line 594: Correct the typo in Prof. Menrad’s name.
Response 19:
The authors have amended the spelling.
Reviewer 2 Report
Comments and Suggestions for Authors
The manuscript entitled ‘Toward National Guidelines for Biodegradable and Compostable Bioplastics: A Case Study in Federal Territory Kuala Lumpur, Malaysia’ has several drawbacks that need to be carefully revised before consideration in the journal.
- Line 32-39: Why is it placed here?. Hedging ‘Introduction is missing’.
- Introduction: Add a brief note on different types of bioplastics in commercial use.
- Why is Table 1 presented in another color?
- Section 3.4: There is no description about the proposed objectives.
- There are several parts in the case study that need more description.
- Fig 2 and Fig 3 are not diagrammatic representations, it is better to change them as text.
- Discussion section should be more elaborated
- There are several grammatical errors the author and team should proof read the entire manuscript.
Author Response
All revisions have been indicated in blue for ease of review
Comment 1: Line 32–39: Why is it placed here? Hedging — ‘Introduction is missing’.
Response 1: Line 35: The authors have added ‘Introduction’
Comment 2: Introduction: Add a brief note on different types of bioplastics in commercial use.
Response 2: Line 36-55: The authors have added a brief note on types of bioplastics
Comment 3: Why is Table 1 presented in another color?
Response 3: It is a response to the editor.
Comment 4: Section 3.4: There is no description about the proposed objectives.
Response 4: Line 228: The authors have revised Table 3 and added the proposed draft.
Comment 5: There are several parts in the case study that need more description.
Response 5: The authors have revised and added descriptions throughout the article
Comment 6: Fig. 2 and Fig. 3 are not diagrammatic representations — it is better to change them to text.
Response 6: The authors have removed the figures and changed them to text
Comment 7: The Discussion section should be more elaborated.
Response 7: Line 685- 693: The authors have added SWOT analysis.
Comment 8: There are several grammatical errors — the author and team should proofread the entire manuscript.
Response 8: The authors have proofread the entire manuscript.
Round 2
Reviewer 1 Report
Comments and Suggestions for Authors
Dear authors, thanks for modifying and amending your publication. The revised version more clearly presents the background and your intentions for publication. I am still unsure, whether the article fits into the scope of 'Polymers', but I leave this decision to the editors.
I am still not satisfied with the introductory definition of bioplastics (lines 36-42). The sentence "Additionally, there are fossil-based bioplastics, which are derived from petroleum." can easily be misunderstood in a way that any fossil-based plastic may be called bioplastic. The point is that ou need the quote 'Bioplastics are bio-based or biodegradable or both' first. Following, there are bio-based plastics on one side. They may be degradable or not - both types may be called bioplastics. On the other side, there are fossil-based plastics. But only such fossil-based materials that are certified biodegradable under defined time and environment belong to bioplastics. I still ask you to be more exact in presenting the general definition. Lateron (lines 116-119), you clarify correctly that the article focuses on bio-based, biodegradable and compostable bioplastics. Please delete the previous statment given in lines 95-96 ('This article focuses on biodegradable and compostable bioplastics.') in order to avoid confusing the readers.
Author Response
Thank you very much for your time and valuable feedback on our manuscript. Please find below our detailed, point-by-point responses to the reviewers’ comments, with corresponding revisions clearly indicated in the re-submitted files. For ease of reference, all changes have been highlighted in blue. Kindly note that any sections or content which the reviewers recommended for removal have been fully deleted from the manuscript and therefore do not appear with highlighted changes.

Reviewer 2 Report
Comments and Suggestions for Authors
Suggested modifications are incorporated by the author and team. The manuscript may be considered for publication.
Author Response

(The authors gave the same response as above.)
